# Subacute Sclerosing Panencephalitis in Children: The Archetype of Non-Vaccination

**DOI:** 10.3390/v14040733

**Published:** 2022-03-31

**Authors:** Laura Papetti, Maria Elisa Amodeo, Letizia Sabatini, Melissa Baggieri, Alessandro Capuano, Federica Graziola, Antonella Marchi, Paola Bucci, Emilio D’Ugo, Maedeh Kojouri, Silvia Gioacchini, Carlo Efisio Marras, Carlotta Ginevra Nucci, Fabiana Ursitti, Giorgia Sforza, Michela Ada Noris Ferilli, Gabriele Monte, Romina Moavero, Federico Vigevano, Massimiliano Valeriani, Fabio Magurano

**Affiliations:** 1Neurology Unit, Department of Neuroscience, Bambino Gesù Children Hospital, IRCCS, 00165 Rome, Italy; alessandro.capuano@opbg.net (A.C.); federica.graziola@opbg.net (F.G.); fabiana.ursitti@opbg.net (F.U.); giorgia.sforza@opbg.net (G.S.); michela.ferilli@opbg.net (M.A.N.F.); gabriele.monte@opbg.net (G.M.); federico.vigevano@opbg.net (F.V.); massimiliano.valeriani@opbg.net (M.V.); 2Department of Pediatrics, Bambino Gesù Children Hospital, IRCCS, 00165 Rome, Italy; mariaelisa.amodeo@opbg.net (M.E.A.); letizia.sabatini@opbg.net (L.S.); 3Department of System Medicine, Tor Vergata University of Rome, Viale Oxford 81, 00133 Roma, Italy; romina.moavero@opbg.net; 4National Measles Reference Laboratory—WHO/LabNet, Department of Infectious Diseases—Istituto Superiore di Sanità (ISS), 00165 Rome, Italy; melissa.baggieri@iss.it (M.B.); antonella.marchi@iss.it (A.M.); paola.bucci@iss.it (P.B.); Emilio.dugo@iss.it (E.D.); Maedeh.Kojouri@iss.it (M.K.); silvia.gioacchini@opbg.net (S.G.); 5Unit of Neurosurgery, Department of Neurosciences, Bambino Gesù Children’s Hospital, IRCCS, 00165 Rome, Italy; carloefisio.marras@opbg.net (C.E.M.); cginevra.nucci@opbg.net (C.G.N.); 6Child Neurology and Psychiatry Unit, Department of System Medicine, Tor Vergata University of Rome, Viale Oxford 81, 00133 Rome, Italy

**Keywords:** measles, subacute sclerosing panencephalitis, treatment, vaccination

## Abstract

Subacute sclerosing panencephalitis (SSPE) is a late complication of measles virus infection that occurs in previously healthy children. This disease has no specific cure and is associated with a high degree of disability and mortality. In recent years, there has been an increase in its incidence in relation to a reduction in vaccination adherence, accentuated by the COVID-19 pandemic. In this article, we take stock of the current evidence on SSPE and report our personal clinical experience. We emphasise that, to date, the only effective protection strategy against this disease is vaccination against the measles virus.

## 1. Introduction

Subacute sclerosing panencephalitis (SSPE) is a rare progressive degenerative disorder of the central nervous system (CNS) caused by persistent measles virus (MeV) infection. Primarily, it affects children and young adolescents and can be fatal [1].

The SSPE burden reflects the epidemiology of natural MeV infection, and it is inversely related to vaccination coverage. According to analyses from the USA and the UK, 4 to 11 cases of SSPE are expected per 100,000 cases of MeV, but the incidence is higher when the infection is contracted under the age of 1 year (18/100,000) [1]. The worldwide incidence of SSPE varies greatly, from approximately 0.2 to 40 cases per million population per year, considering geographical disparity, with a reduction of 82–96% in countries that have achieved increased vaccination coverage, in particular from 2000 to 2016 [2]. However, despite the Global Vaccine Action Plan endorsed by the World Health Organization (WHO), the global elimination of MeV has taken a significant step backward recently, with an increase in cases to 120 per million in 2019, the highest since 2001. Vaccine refusal, an emerging and spreading trend in the population, is considered a significant reason for these changes in population-level susceptibility to MeV [3]. Moreover, the COVID-19 pandemic has further slowed vaccination and surveillance programs [4]. During the international containment measures, reports showed measles–mumps–rubella vaccination counts to be about 19.8% lower in 2020 compared to the same period in 2019 [5].

Given this dangerous scenario, new future outbreaks of MeV infection and related complications are possible if appropriate public health programs are not implemented with immediate effect [4].

Among long-term complications of MeV infection, SSPE has the longest latency, varying from 1 month to 27 years, with a median age at diagnosis of 12 years [6]. A shorter latency is related to an early primary infection, at under 2 years of age, usually associated with family clusters, mainly due to the lack of parental vaccination coverage. Although no gender predisposition has ever been proven, a higher incidence has mainly been reported in boys, with a male-to-female ratio of 2.8:1 [7,8].

After primary infection, persistent virus replication in the CNS leads to progressive destruction of the neurons, which becomes clinically evident several years later [9,10]. Neurons are not the only target of the virus, and numerous neuropathological studies on the brain tissue of SSPE patients have established that, apart from the neurons in the grey matter, many oligodendrocytes are infected in the white matter [11,12,13]. Astrocytes are also infected throughout the CNS, but to a much lesser degree [13].

### 1.1. Pathogenesis

The pathogenesis of the disease is still under investigation. SSPE is caused by hypermutated MeVs combined with an inadequate cellular response [14].

Although the MeV is serologically monotypic, genetic variability has defined eight clades, including 23 genotypes and a putative new genotype, that are geographically and temporally restricted [15]. Despite the paucity of studies on the molecular epidemiology of SSPE, MeV sequences obtained from brain tissues are homologous to the genotype circulating at the time of primary exposure to MeV [16]. There is no evidence to indicate that wild-type MeV strains differ in terms of either pathogenesis or neurovirulence [17]. Studies on brain tissue of SSPE patients have shown that the MeV in these patients has several genetic mutations that enable the virus to spread and persist in the human brain [18,19,20]. Since human neurons, an important target affected in the disease, do not express the known MeV receptors (signalling lymphocyte activation molecule (SLAM) and nectin 4), how the MeV infects neurons and spreads between them is unknown [21].

MeV recovered from patients with SSPE differs from wild-type viruses in mutations involving the matrix (M), hemagglutinin (H), nucleocapsid (N) and fusion (F) genes [22]. The MeV genome is encapsidated by the N protein and forms the ribonucleoprotein (RNP) complex with the viral RNA-dependent RNA polymerase comprised of the L and P proteins. The two envelope glycoproteins have a role in receptor binding (H protein) and membrane fusion (F protein) [22]. The M gene is highly mutated in almost all cases in the entire open reading frame [18,19]; the N is modified in the carboxyl terminus, and the H has biased hypermutation in a limited region [15,17]. Further, many persisting MeVs have mutations in the F gene, which cause the cytoplasmic tail of the F protein to elongate or shorten [23,24].

Mutations in the M protein impair the formation of new viral particles, helping the replicating virus to persist in the neuronal cells, spread through the synapses and evade the neutralising antibodies [6]. Studies have detected a particular triresidue structural motif at residues 64, 89 and 209, with proline, glutamate and alanine at these positions, respectively, (called PEA) in the primary sequence of the SSPE M proteins, which is absent in vaccine and lab-adapted strains, confirming the theory that only the wild-type virus can cause SSPE [25].

Recent studies have revealed that changes in the ectodomain of the MeV F protein play a key role in the MeV spread in the brain [26,27,28]. Changes in the F protein render it hyperfusogenic, allowing the virus to propagate in the neurons [21,29]. Notably, the F protein, not the receptor-binding H protein, changes during persistence to allow the virus to exhibit tropism for neurons [22]. The changes in the M protein and the cytoplasmic tail of the F protein affect the interaction between the two proteins, increase the surface expression level of the F protein and enhance cell-to-cell fusion [30]. Hyperfusogenic F proteins permit MeVs to enter cells and spread without the need to engage nectin-4 or CD150, known receptors for MeV not present on neural cells [31]. Changes in the H and F proteins can also be associated with persistent infection, with the M protein remaining relatively unaffected [24,32]. Since all three proteins are associated with viral budding from infected cells and putative fusion with uninfected cells, the infection is thought to persist due to defects in these two processes [6,33].

Another hypothesis is that the MeV acts upon another cell receptor heavily concentrated at the synapses to gain entry into a neuron [34]. One of the proposed mechanisms of neuronal spread is via neurokinin-1 [35], while substance P and the fusion inhibitory peptide block viral transmission [22]. CD9 may also play a role in this process because high levels of antibodies against it have been found in the cerebrospinal fluid (CSF) of individuals with SSPE and severe brain atrophy [36].

Some studies have shown that cell adhesion molecule 1 (CADM1) and CADM2 are host factors that enable the MeV to cause membrane fusion in cells lacking the known receptors and to spread between neurons [21,37]. CADM1 and CADM2 interact in cis with the H protein on the same cell membrane, triggering hyperfusogenic F protein-mediated membrane fusion [37]. Recently, a process has been discovered involving transmembrane and cytoplasmic protein transfer that relies on cell–cell contacts established by the nectin adhesive interface. This process has been called nectin-elicited cytoplasm transfer (NECT). NECT can spread MeV infections from epithelial cells to primary neurons, which is possibly the first step of neuropathology [38]. However, the role of CADM1, CADM2 and NECT in favouring the spread of the virus in the brain tissue must be verified.

Once inside the cell, the MeV changes the cell machinery to bypass the immune system and continues reproducing itself inside the cell in a less cytopathic fashion to avoid destroying the host neurons [39].

SSPE may develop only in the case of a particular susceptibility of the host [6]. Impairment of determined immune system functions has been suggested as a leading cause of that susceptibility. Some Japanese studies have demonstrated that mutations in three genes, MxA (an antiviral protein induced by interferon-alpha and interferon-beta), interleukin-4 and interferon-1, are associated with a high risk of SSPE after MeV infection [40,41]. Other studies have suggested a deficient interferon (IFN-α) system in SSPE, with an imbalance in the production of IFN-α in response to stimulations in vitro [42]. Affected individuals showed an imbalance in the high levels of cytokines promoted by Th2 cells, such as IL-4 and IL-1b, and the low levels of cytokines promoted by Th1 cells, such as IFN-α, IL-2, IL-10 and IL-12 [42,43]. This cytokine storm enhances the humoral instead of the cytotoxic response, leading to a predisposition to the self-replication of the virus in the brain [41]. There is also a hypothesis suggesting that SSPE could affect people with transient immunosuppression due to an exposure to another pathogen at the moment of MeV primary infection [44].

Transmission among neurons is mediated by neurokinin synaptic receptors and genetic polymorphism favouring either a humoral response or intraneuronal spreading due to the facilitating polymorphism of the entry molecules [35].

### 1.2. Clinical Features and Diagnosis

A clinical picture of SSPE is characterised by an insidious and subacute cognitive decline and behavioural alterations, followed by movement disorders, such as myoclonus and ataxia, seizures and visual impairment. After the initial presentation, the disease can progress rapidly, leading to a vegetative state or death within a few years. Theoretically, the clinical course is divided into four stages, according to Jabbour et al. [45], summarised in Table 1.

Atypical presentations, accounting for 10% of all cases, consist of isolated psychiatric manifestations, gait disturbance as an initial symptom, poorly controlled seizures, isolated extra pyramidal symptoms, visual loss and/or lateralising motor weakness. Usually, these patients do not follow any defined clinical course of the disease.

SSPE can be established according to Dyken’s criteria, described in the Materials and Methods section [46].

An electroencephalogram (EEG) is a valid tool to support the diagnosis. The typical EEG finding consists of periodic, generalised, symmetrical, stereotyped complexes, with high-voltage diphasic waves occurring synchronously throughout the recording and usually clinically associated with myoclonic jerks. These complexes are pathognomonic features of SSPE.

Neuroimaging can provide useful information but is not mandatory for diagnosis. One of the most common features in the magnetic resonance (MR) study of the brain is hyperintensities on T2-weighted images in the cerebral cortex, the periventricular white matter, the basal ganglia and the brainstem, present in the medium stage of the disease. Along with disease progression, diffuse cortical atrophy may be revealed.

The diagnosis can be confirmed by the detection of MeV antigens by a brain biopsy using a reverse-transcriptase polymerase chain reaction for MeV RNA [47].

### 1.3. Treatment

Currently, there is no specific validated therapy for SSPE. Nevertheless, considering the fatal course of the disease and the possibility of reaching a satisfactory outcome in up to 35% of patients who receive treatment in terms of clinical improvement, clinical stabilisation, slowdown of progression and prolonged survival, it is mandatory to treat patients with SSPE [39].

The most reliable therapies include inosiplex, ribavirin and interferon in different formulations and doses, even if the efficacy of these treatment approaches has not been clearly established (Table 2).

Inosine pranobex (IP), a derivative of inosine and the *p*-acetamido-benzoic acid salt of N,N-dimethylamido-2-propanol, has been used since around 1971 for its immunomodulatory and antiviral properties with a safety profile. It stimulates the proliferation of T-cell lymphocytes, the functioning of natural killer cells, the release of pro-inflammatory cytokines and the inhibition of viral RNA synthesis. The inosiplex dosage was 100 mg/kg/day to a maximum of 3 g/day, taken orally in three divided doses for 6 months [58].

Ribavirin is a broad-spectrum antiviral agent that inhibits a wide range of RNA and DNA viruses. Considering the difficulty in crossing the blood–brain barrier, its efficacy in treating SSPE was discussed, and a new option of intraventricular administration was recently proposed [59].

Other antiviral drugs, such as favipiravir, have been shown to inhibit RNA viruses only in vitro, but this needs further investigation [60].

Interferon therapy (IFN-α) seems to be a promising treatment produced by recombinant DNA techniques, based on the hypothesis that MeV persists on the SNC due to the humoral response instead of a cellular physiological immune response, mediated by IFN-α.

IFN-α therapy also seems to have a positive effect on maintaining the metabolism of the cerebral cortex, as demonstrated through a serial ^1^⁸FDG-PET study on a 15-year-old girl with stage II subacute sclerosing panencephalitis (SSPE) treated with isoprinosine, IFN-α and ribavirin for 3 years [61].

Despite the lack of clinical studies on the improved results for treatment with IFN-α combined with another drug compared with only inosiplex monotherapy, the combination strategy was suggested [54].

This theory is based on preclinical research giving evidence that the combination of IFN-α with other drugs, such as ribavirin or inosiplex, should be preferred because of a synergic inhibitory effect on the virus’s replication in vitro and in hamster models [62].

In a clinical study in the 1990s, an improvement in the survival curves of patients treated with INF-α compared with control cases was found only in the first 8–9 years. A positive clinical effect of therapy was observed in 72% of the patients in the initial phase of the follow-up. After a period of 6 to 60 months without therapy, the clinical conditions deteriorated in most patients. Therefore, it was suggested that the long-term administration of combined therapy with inosiplex is more effective than its short-term administration [43].

## 2. Materials and Methods

### 2.1. Dyken’s Criteria for SSPE Diagnosis

Dyken’s criteria are divided into two major and four minor criteria, as described in Table 3:

### 2.2. Case Descriptions

In the present study, we describe the clinical presentation and the laboratory and instrumental findings for four paediatric cases affected by SSPE, each of whom underwent a long-term combined intrathecal IFN-α therapy. The main aim is to analyse the clinical course of the disease and the therapeutic outcome during a long-term follow-up.

This is a case series of paediatric SSPE patients monitored at the Bambino Gesù Children’s Hospital (Rome, Italy) over 10 years (2011–2021). We retrospectively enrolled 4 patients, all boys, who met Dyken’s criteria for the diagnosis of SSPE. We collected personal information, medical histories and laboratory data from electronic medical records. All patients underwent haematological investigations, including autoimmune workup, MRI, cerebrospinal fluid (CSF) study and EEG at the time of diagnosis. Measles antibodies were measured in the CSF and serum using enzyme immunoassays (ELISA). Then, CSF and serum specimens were sent to the WHO/LabNet National Reference Laboratory (NRL) for measles and rubella, where the level of the neutralising antibody was investigated by the plaque reduction neutralisation test (PRNT80) [63]. Molecular detection of the measles genome was attempted by means of RT-PCR, as previously described [64]. Sequencing of the N-450 region was performed on a measles-positive sample to establish the strain responsible for the infection.

All the patients were treated with combined intrathecal IFN-α therapy and this was followed up by a neurological examination, EEG and MRI at specific intervals after the initial reports. Restaging was made according to standardised classification at 6 months after diagnosis and during the last medical check-up.

## 3. Results

### Case Series

We presented four male children, three of whom were exposed to measles before the first dose of vaccine, between 14 days and 11 months of age, with a confirmed maternal epidemiological link. Only one of them was exposed to the virus at 3 years of age, when he had already received the first dose of the measles vaccine, without a clear link of infection. In particular, the parents reported an episode of fever with a mild skin rash lasting 3 days.

In the same patient, it was hypothesised that a single dose of the vaccine did not result in sufficient immunological protection against primary infection, although immunological congenital defects were excluded with specific investigations (white blood cell count; lymphocyte subpopulation count; immunoglobulin Ig G, Ig M and IgG subclasses dosage and study of the lymphocytic phenotype B; and specific vaccine response to *Haemophilus* and *Pneumoccoccus vaccinatiom*). No information about the response to the first dose of the measles-containing vaccine (MMR) before the onset of neurological symptoms was found in the medical records, but the patient showed anti-measles IgG antibodies in his blood and liquor at the onset of SSPE symptoms.

Clinical, instrumental and laboratory findings are provided in Table 4.

None of the patients received infusion of polyvalent immunoglobulin for post-exposure prophylaxis. None received a dose of vaccine prior to infection with MMR. Three of the patients had a similar latency time between exposure and SSPE onset, with a mean standard deviation (SD) age of 4.1 years ± 0.42 (range 46.30–55 months). The older infected boy had an increased latency of 12 years. 

Clinical presentations at the onset ranged from stage 2A to stage 3 and included a wide spectrum of insidious symptoms, from abnormal behaviour and recurrent falls to visible motor and cognitive impairment. Children were referred to our hospital with a clear neurological symptomatology, and all of them presented myoclonus and atonic seizures. In addition, ballistic movements of the lower limbs and atypical clinical history with poorly controlled seizures were observed in patient B. Only one of them, patient C at 6 years of age, presented a rapid global neurological deterioration with spastic quadriplegia during an infective febrile episode. The same patient showed the better response to therapy in clinical terms, improving from the clinical stage 3 to stage 2D.

Diagnosis was made according to Dyken’s criteria. All patients fulfilled 3 out of the 4 criteria, in particular, clinical history, typical EEG and elevated CSF globulin levels, with the exception of the invasive brain immunopathology assay, which was not performed. Figure 1 and Figure 2 present the typical EEG findings for SSPE.

The EEG showed a typical periodic pattern in all patients and was a main exam used to guide studies by other laboratories (Figure 1 and Figure 2).

T2-weighted sequences of brain MRI showed hyperintensities in all patients, but in different areas, apparently not related to clinical severity (Figure 3).

Blood examinations at diagnosis showed a normal blood cell count and normal markers of inflammation. All patients had high levels of anti-measles IgG in the CSF and in blood, while patient D also presented anti-measles IgM in blood and in the CSF.

Normally, IgG is not produced intrathecally in any significant amount and most of the IgG found in the CSF is derived peripherally and crosses the blood–brain barrier (BBB). The presence of IgG in the CSF is, therefore, not synonymous with CNS inflammation. However, in a range of neuroinflammatory conditions, most notably MS, a small number of B cell clones in the CNS produce oligoclonal IgG.

Although the serological profile shows a previous measles infection in the absence of vaccination, the high measles IgG titre suggests reinfection or reactivation of the virus.

As shown in Table 5, all the patients had high levels of neutralising antibodies against measles, in both serum and the CSF, quantified by PNRT80 assay.

No measles genome was detected in the blood or CSF specimens, apart from patient C. In this case, the genome was detected in the CSF sample and sequenced for genetic characterisation. The strain responsible for the previous measles infection was found to belong to the MeV genotype B3 and to be 99.8% similar to strains circulating in Italy in 2011 and closely related to the strain named MVi/Harare.ZWE/38.09/1 (MeaNS ID 11292), the year of the suspected exposure. The genome sequence was submitted to the Measles Nucleotide Surveillance database (MeaNS ID 151625).

Brain biopsy was not required for diagnosis in any of our patients as they met the remaining major or minor criteria.

The treatment regimen consisted of IFN-α intrathecal infusions through a Rackham reservoir, started at 100,000 units/m^2^ and increased to 1 million units/m^2^ of body surface area (BSA) per day within 5 days; after allowing the patients 2 days’ rest, we administered a dosage of 1 million units/m^2^ of BSA weekly for 6 months, followed by monthly infusions (Table 6). Patient B was treated with a continuous intrathecal infusion pump after a 5-day regimen loading dose and 6 months of monthly infusion. Three patients were also given oral inosiplex 100 mg/kg daily, while one of them, patient D, received oral ribavirin 20 mg/kg daily for 7 days and an infusion of IVIG and vitamin A 50,000 UI two times a day. Symptomatic neurological therapy was added to control seizures and myoclonus. Patient D received vitamin A supplementation despite the absence of signs suggestive of vitamin deficiency (e.g., xerophthalmia, Bitot’s spots and corneal ulceration), in accordance with the most recent guidelines that suggest this approach in the case of severe and potentially life-threatening complications of MeV infection [14].

IFN-α therapy was administered for a period ranging from 6 months to 3 years, while adjuvant therapy was continued from 3 months to 3.6 years. Patient D had been recently diagnosed, and after 9 months of therapy he discontinued treatment due to poor family compliance.

As far as side effects related to IFN-α therapy are concerned, we registered fever, lethargy and vomiting during the period 24–48 h after the first IFN-α infusion in 3 out of 4 patients. The fever responded to paracetamol and rapidly disappeared, while vomiting was mild and resolved spontaneously. A side effect related to the Rackham device was documented only in patient D, who was admitted to our department after 3 months of therapy because of a cerebral abscess in the same area as the intraventricular device, which was promptly removed and repositioned. A methicillin-resistant *Staphylococcus aureus* strain was isolated in the patient, but he improved in terms of clinical conditions with the administration of a wide-spectrum intravenous antibiotic therapy. Three months later, the same therapy regimen was re-administered because of a rapid neurological deterioration.

In the last follow-up, at the age of 12 years, and after 4 years without therapy, patient C was diagnosed with SARS-CoV-2 infection, Omicron variant. His clinical conditions were good, with fever for 4 days, oxygen saturation always at about 99% and no other symptoms. He was administered one dose of monoclonal antibody Sotrovimab 250 mg.

The clinical response to therapy was different in our patients: patient A showed a slow progression, patient B stabilised, patient C improved and patient D showed a progression, probably related to the cerebral abscess, which delayed therapy (Figure 4).

## 4. Discussion

The neurological sequelae of primary measles encephalitis (PME), acute post-infectious measles encephalomyelitis (APME), measles inclusion body encephalitis (MIBE) and SSPE post MeV are less common than other MeV-related complications and can lead to severe disability or death.

Diagnosis of MeV-related SSPE is based on the characteristic symptomatology, specific electroencephalographic changes and elevated titres of MeV antibodies in the serum and the CSF.

We presented a group of SSPE patients who underwent long-term treatment with INF-α intrathecal 1,000,000 U/m^2^ once a week and isoprinosine 100 mg/kg/day orally for a mean period of 24 months.

This treatment scheme is the one suggested by the International Consortium on SSPE [54].

Our case series reports an improved response with treatment for a longer duration with the combination of intrathecal INF-α and oral isoprinosine. However, we did not find different results compared to those of the trial by Gascon et al. [54]. In the work of the international consensus, compared to treatment with isoprinosine alone, the combined therapy used for 6 months did not show significant efficacy in terms of morbidity outcomes (improvement, stabilisation, worsening after treatment stopped, deterioration). In the first 6 months of follow-up of our case series, we observed two good clinical responses (patients B and C), confirmed in the last control, in terms of clinical stabilisation (1 year and 4 years without therapy, respectively). These patients were treated for a longer period, of about 3 years, without therapy discontinuation.

Furthermore, the clinical picture showed stability in children who had stopped treatment. Overall, in the last follow-up, ranging from 1 year to 7 years after starting therapy, the best result we detected was a slowdown in disease progression in three out of four patients. Only one patient showed a rapid clinical progression, with neurological deterioration, probably related to the complication of a cerebral abscess causing a discontinuation in therapy. Nevertheless, even when we found a positive response in terms of slow progression or life extension, the quality of life remained low due to the high rate of neurological and neurovegetative morbidity that characterises the disease.

In the recent literature, there is one case report describing the longest follow-up, of 13 years, in a female patient aged 13 years at diagnosis and treated with oral inosiplex for 5 years and intraventricular INF-α for 13 years. The complex positive effect in terms of clinical improvement was mentioned, without many side effects except for one episode of bacterial meningitis [66]. A more detailed study analysed 18 patients treated with at least 12 months of intraventricular INF-α and oral inosiplex, reporting clinical stabilisation or improvement in 44% of the patients [57].

Taking into account the studies available, IFN-α therapy was conducted for a period no longer than 3–12 months, improving clinical conditions in 10% to 36% of the cases. The therapy was administrated in different ways, such as intra-muscular, subcutaneous and intravenous. The follow-up ranged from 6 months to 3 years, with the exception of the above-mentioned case report [54,56,59,67].

Compared with previous case reports on the use of intrathecal INF-α and orally isoprinosine, our study showed that longer treatment duration does not offer improved results except in terms of an increase in survival and disease stabilisation.

Regarding other therapy schemes, a previous study described the treatment of three patients with combined oral isoprinosine and intrathecal ribavirin plus INF-α. The treatment lasted more than 5 years. In this case, the authors found a slowdown in progression in two patients and stability in one patient.

Even early treatment does not lead to a reduction in the stage of the disease. In fact, our patients who started treatment within a few weeks of diagnosis showed final staging levels associated with high disability and morbidity. Already highlighted by other authors, this often leads to younger children with a rapidly progressive disease course from onset [57].

The data in the literature are conflicting regarding the effects of the various treatment protocols used [50,51,53]. In particular, controlled studies are scarce and highlight how the long-term use of antivirals or immunomodulators can be considered in patients with SSPE to obtain benefits in terms of stabilising the disease or increasing survival [51,53,55].

With the exception of data from the international consortium on SSPE, the available data in the literature are contrasting and based on small case series, with different combinations of drugs used and different treatment durations. In particular, little information is available for the paediatric population about a protracted therapy with IFN-α in terms of efficacy and side effects.

WHO recognises vitamin A deficiency as a risk factor for MeV [68]. The effect of vitamin A in MeV may be mediated by the integrity of epithelial cells and the reinforcement of the immune system [69]. However, the role of vitamin A in the pathogenesis of SSPE needs to be clarified, as there are no studies demonstrating efficacy in the treatment of the disease. Only one of our patients (patient D) underwent treatment with vitamin A according to the WHO scheme but without presenting positive changes in the disease stroke compared to the others.

SSPE is a serious disease that, to date, does not have a specific cure capable of leading to a regression of the symptoms or preventing associated disability [70,71]. As of now, vaccination is the only valid weapon to prevent the disease and it should be considered as both an individual and a social duty. In addition, individual patients may have different backgrounds, modalities of onset and clinical courses. This diversity contributes to making diagnosis and treatment of SSPE patients difficult.

Since the MeV vaccine is approved from 12 months of life, the first year is the period the patient is most at risk of a possible infection because of the absence of immunity. In fact, three out of the four cases we presented contracted MeV infection during this time window, when relatives’ vaccination coverage is a matter of major concern. In these cases, SSPE develops with a shorter latency, in line with data in the literature [6]. Moreover, there is a lack of evidence and standardised guidelines about postexposure prophylaxis in children without evidence of immunity. None of our cases underwent immune globulin administration after exposure, which may prevent or modify the clinical course of MeV according to observational studies and meta-analyses [43,44,45].

A recent literature review [72] analysed the current positions as set out in the national guidelines of the United States, Australia, New Zealand, Canada and the United Kingdom, concluding that intravenous immunoglobulin (IVIG) or intra-muscular immunoglobulin (IMIG) could be administered to infants aged less than 6 months as early as possible after exposure.

A systematic review [73] analysed the role of passive immunisation within 7 days of exposure in non-vaccinated people to prevent MeV infection and concluded that treated people had 83% less risk of primary infection than non-treated people. Interestingly, there is no systematic analysis in the literature of the efficacy of post-exposure prophylaxis to prevent long-term complications from MeV, such as SSPE [74,75].

Endo et al. [76], assessing the efficacy of postexposure prophylaxis against MeV with immunoglobulin, found a titre-dependent effect, with higher anti-MeV titres providing the greatest protection. Children who did not develop the disease had received a mean dose of 10.9 IU/kg compared with 5.7 IU/kg for children in whose case post-exposure prophylaxis with IG intramuscular failed.

In none of the SSPE series present in the literature we have data on whether post-exposure prophylaxis prevents SSPE. Data in this area would help to determine whether this could be a valid strategy to reduce the risk of short- and long-term complications associated with MeV. This is an important issue, especially in the COVID-19 era, if we consider that during the pandemic and the lockdown periods, worldwide, there has been a reduction in vaccination campaign compliance, with an increase in the cases of MeV in the paediatric population and therefore a risk that in the near future more SSPE cases will be seen [4,5]

Diagnosis of SSPE is challenging and relies on the detection of increased concentrations of MeV IgG antibodies in the cerebrospinal fluid in a clinically compatible context. The results from serological testing alone are not convincing proof of intrathecal production of MeV IgG. MeV viral copies in the blood or the CSF may be absent [14].

Intrathecal antibody synthesis against the MeV causative antigen must be detected to confirm a diagnosis of SSPE [77]. In this view, the methodology of the exam is relevant because qualitative methods, such as the antibody index (AI), can help detect the intrathecal antibody for a causative antigen as well as a part of a polyspecific immune response. The quantitation of the intrathecal antibody fraction in the CSF with ELISA helps to discriminate both cases [78].

Just the presence of anti-MeV IgG in blood is by no means the basis for a diagnosis because it could be due to a previous vaccination, as, for example, in our patient B. Characteristically, a child with SSPE is one who was always healthy, who contracted or had exposure to a case of MeV and who after a few years suddenly and progressively developed a neurological deterioration with atonic epileptic seizures. In these children, the EEG with typical periodic complexes already adds a strong suspicion of SSPE. In these patients, it is mandatory to perform a lumbar puncture to determine anti-MeV Ig in the CSF. A possible aim of future SPPE research would be to investigate whether children who had contracted MeV prior to vaccination could then be screened for the risk of developing SSPE by studying the susceptibility genes (MxA, interleukin-4 and interferon genes, INF-α) we discussed above.

The lack of a blood marker also supports the hypothesis that SSPE is associated with an inflammation state localised only in the CNS, giving a reason for intrathecal target therapy. Results from the genetic analysis of the MeV genome in one of the reported cases confirm that the strain identified was similar to the strains that circulated at the moment of infection.

To the best of our knowledge, this is the first case report on SSPE with long therapy courses and the longest follow-up.

## 5. Conclusions

In conclusion, many steps are still needed to achieve the goal of eradicating measles. Considering the insufficient data on post-exposure prophylaxis efficacy and the limitations of available therapies, the only concrete strategy against measles and related complications is prevention through vaccination. The COVID-19 pandemic has led to dangerous immunity gaps resulting from suspended awareness campaigns and delayed immunisation activities, leading to medical concern over the risk of new measles outbreaks in the coming years. The relentless and fatal course of SSPE that we have highlighted in this study underscores the importance of vaccination programs and demands urgent action to reverse a pending measles catastrophe that will affect children and public health.

## Figures and Tables

**Figure 1 viruses-14-00733-f001:**
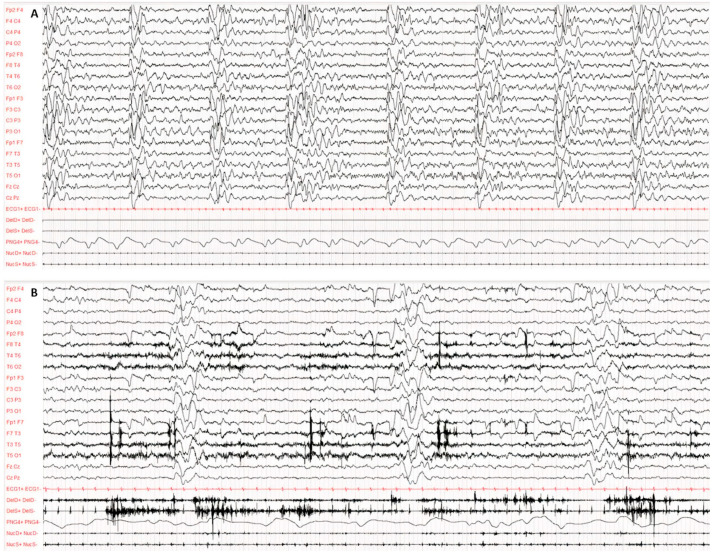
Typical EEG at the onset of symptoms, performed according to the international 10–20 system with digital acquisition and polygraphy. EEG recording of 60 s awake (**A**) and during sleep (**B**). Pathological background activity is globally slowed down, disorganised and undifferentiated both in wakefulness and in sleep. Continuous periodic (about 5–6 s) and polyphasic complexes consisting of several high-voltage, irregular and bilateral asynchronous delta waves. The polygraphy recorded atonic seizures of limbs, head and face concomitantly with the periodic complexes.

**Figure 2 viruses-14-00733-f002:**
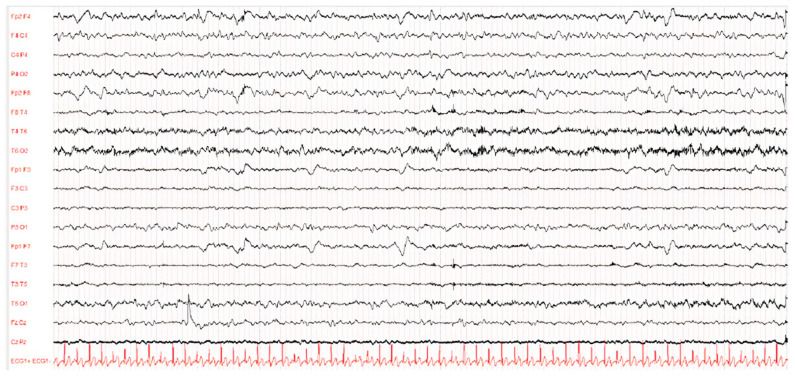
EEG of patient D after 1 year of illness. The waking EEG trace showed globally disorganised, undifferentiated and asymmetrical brain activity due to the presence of hypovolted activity in the left frontal–central–temporal regions.

**Figure 3 viruses-14-00733-f003:**
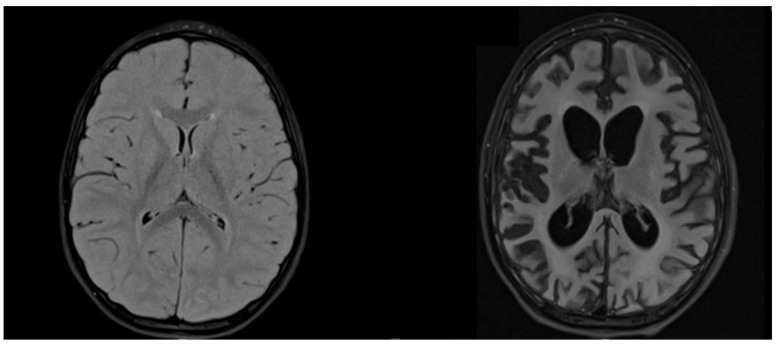
MRI of patient D. Axial T2-weighted FLAIR MRIs at the time of diagnosis (age 3.9 years) and after 1 year of illness (age 4.9 years) show, respectively, signal alteration of the bilateral posterior parieto-occipital region and severe cerebral atrophy.

**Figure 4 viruses-14-00733-f004:**
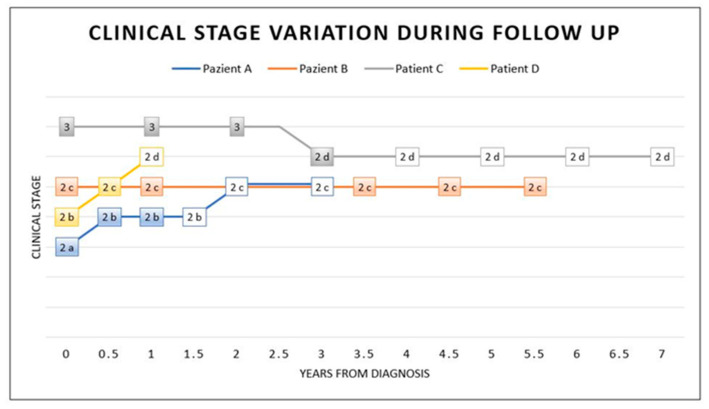
Stage variation during and after treatment. This graphic shows the clinical stage variation at diagnosis (time 0), during IFN-α therapy (coloured box) and after therapy (white box). Intrathecal IFN-α therapy was considered the main treatment in this graphic (coloured box). As illustrated, at time 0, a diagnosis was made and coincided with the start of the treatment. Patient A took therapy for 1 year. He presented mild clinical progression in the following year, with relative stabilisation in stage 2c, even after 2 years of therapy discontinuation. Patients undergoing IFN-α therapy for a long time (5.5 years for patient B and 3 years for patient C) showed clinical stabilisation (patient B) or a mild improvement with a subsequent stabilisation even after 4 years without therapy (patient C). Patient D took therapy for a short time, with a discontinuation, and he showed clinical progression.

**Table 1 viruses-14-00733-t001:** Clinical stages of SSPE according to Jabbour et al. [45].

Stage	Clinical Features
**1**	Irritability, personality changes, difficulty in school, lethargy and/or speech impairment
**1A**	Mild mental and/or behavioural changes
**1B**	Marked mental changes
**2**	Movement disorders, such as dyskinesia, dystonia and myoclonus, seizures and/or dementia
**2A**	Myoclonus and/or other involuntary movements and/or epileptic seizures
**2B**	Focal deficits
**2C**	Marked involuntary movements, severe myoclonus and/or focal deficits with impairment of daily activities
**2D**	Akinetic mutism, vegetative state, decerebrated, decorticated rigidity or coma
**3**	Extrapyramidal symptoms, decerebrate posturing and/or spasticity
**4**	Coma, vegetative state, autonomic failure or akinetic mutism

**Table 2 viruses-14-00733-t002:** Data on the treatment of SSPE at various paediatric mean ages. See attached file.

Authors and Type	Location	Number of Subjects	Mean Age (Years)	Therapy	Stage of Disease at the Start of Therapy	Duration of Treatment	Follow-Up Duration	Results	Adverse Effects	Recommendations
[48] Open label	USA	15	11.5	Isoprinosine 100 mg/kg/day	IV (7 sbj) III (1 sbj) II (6 sbj)	22 months (mean)	49 months (mean)	4 died, 4 stable, 7 reduction in stage of 1 point	Mild hyperuricemia	Isoprinosine may be efficacious in the treatment of SSPE.
[49] Clinical trial	USA Canada	98	9.8	Isoprinosine 100 mg/kg/day	NR	1 month–9 years (range)	1 year (mean)	Probability of survival at 2, 4, 6 and 8 years from onset of SSPE was 78%, 69%, 65% and 61%, compared with 38%, 20%, 14% and 8%, respectively, in a composite control group (*p* < 0.001)	NR	Inosiprinosine seems to be able to prolong life in patients with SSPE.
[42] Open label	Israel	11	14.5	Isoprinosine 4 mg	II (10 sbj) I (1 sbj)	3 months–7 years (range)	NR	NR	No side effects	The expected downhill clinical course of SSPE was not influenced by isoprinosine.
[50] Open label	Lebanon	18	9.5	Isoprinosine 50–100 mg/kg/day	IV (3 sbj) III (2 sbj) II (13 sbj)	3–27 weeks (range)	NR	10 died, 3 stable, 5 slow decline	No side effects	There is little support for any clinical efficacy of isoprinosine in SSPE.
[51] Case series	Japan	3	15	Ribavirin 1–4 mg/kg/day and INF-α 6 × 105 IU/day with continuous intraventricular plus isoprinosine 100 mg/kg/day orally	II (2 sbj) III (1 sbj)	Ribavirin plus INF-α for 14 days, repeated after 10–21 days, and daily isoprinosine for 5–7 years	5–7 years (range)	2 slow progression, 1 stable	Swelling of the lips and gums, as well as disturbance of consciousness for bolus administration of ribavirin haemolytic anaemia	The clinical symptoms were temporarily relieved in all cases.
[52] Case series	Japan	5	10.5	Ribavirin 1–9 mg/kg/day and INF-α (dose NR) intraventricular plus isoprinosine (dose NR) orally	IV (2 sbj) III (2 sbj) II (1 sbj)	Ribavirin for 10 days, repeated after 20 days	3–13 months (range)	4 improved, 1 progression	Lip swelling, conjunctival hyperaemia and drowsiness	Intraventricular administration of ribavirin is effective against SSPE if the CSF ribavirin concentration is maintained at a high level.
[53] Case series	Japan	2	13.5	Ribavirin 10–30 mg/kg/day IV for 7 days, combined with intraventricular INF-α therapy (300 × 10^4^ IU 3 times a week) and oral isoprinosine (5600 mg/day)	III (1 sbj) II (1 sbj)	Ribavirin for 7 days, repeated weekly for more than 6 months	NR	1 stable, 1 improved	Reversible anaemia and oral mucosal swelling	Intravenous administration of high-dose ribavirin combined with intraventricular administration of INF-α should be further pursued for their potential use in the therapy of SSPE patients.
[54] RCT	International	67	8.5	Group A: 39 sbj inosiplex 100 mg/kg/day orally Group B: 28 sbj INF-α intrathecal 1,000,000 U/m^2^ twice a week	IIB (27 sbj) IIA (33 sbj) 1B (17 sbj) 1° (3 sbj)	6 months	NR	Group A: 8 died, 2 improved, 10 stabilised, 6 worsened after treatment stopped, 17 deteriorated, 4 insufficient data; Group B: 4 died, 2 improved, 6 stabilised, 1 worsened after treatment stopped, 14 deteriorated, 5 insufficient data	Hyperpyrexia CNS infection, shunt infection and blocked reservoir	There was no statistically significant difference between the two groups on three outcome measures: the Neurological Disability Index, the Brief Assessment Examination and stage.
[55] Case series	Japan	2	13.5	Isoprinosine (180–200 mg/kg/day orally) daily and 300 × 104 IU of intrathecal IFN-α 3 times a week and IV ribavirin with an initial dose of 30–100 mg/kg/day for 1 week	III (1 sbj) II (1 sbj)	3–13 months	NR	2 stable	Hypertemia, anaemia, lip swelling	Early administration of intrathecal high-dose INF-α and IV ribavirin should be considered as a possible therapy for SSPE.
[56] RCT	Turkey	19	5.5	Isoprinosine (100 mg/kg/day orally), lamivudine (10 mg/kg/day orally) and subcutaneous interferon IFN-α 10 mU/m^2^/3 times a week	IIIB (8 sbj) IIIA (3 sbj) IIB (3 sbj) IIA (3 sbj) IB (2 sbj)	IFN-α for 6 months and isoprinosine and lamivudine given during follow-up	16 months (mean)	3 died, 8 worsened, 4 stable, 3 improved	Hypertemia and irritability	Combination treatment protocol resulted in higher remission rates and longer survival periods when compared with controls.
[57] Open label	Saudi Arabia	18	8.9	Oral isoprinosine (100 mg/kg/day) and intraventricular INF-α starting at 500,000 U twice a week and later increased to 3 million U biweekly	II (11 sbj) III (7 sbj)	Oral isoprinosine, 100 mg/kg/day and intraventricular IFN-α beginning at 500,000 U twice a week, increased to 3 million U every 2 weeks	10 months (mean)	4 died, 3 improved, 4 stable, 7 worsened	Ventriculitis-meningitis, thrombocytopenia, febrile reactions and lethargy	Combined oral isoprinosine intraventricular interferon appears to be an effective treatment for SSPE.

**Table 3 viruses-14-00733-t003:** SSPE diagnostic criteria.

Major	
**1**	Elevated CSF measles antibody titres *
**2**	Typical or atypical clinical history
**Minor**	
**3**	Typical EEG
**4**	Elevated CSF globulin levels **
**5**	Brain biopsy
**6**	Molecular diagnostic test to identify the MeV mutated genome

Two major criteria plus one minor are usually needed. If the presentation is atypical, criteria 5 and/or 6 may be required. * Anti-measles antibodies greater than or equal to 1:4 in the CSF. ** More than 20% of the total protein found in the CSF.

**Table 4 viruses-14-00733-t004:** Clinical, instrumental and laboratory findings in children with SSPE at diagnosis.

General Information					Clinical Findings at Diagnosis			CSF Analysis					Instrumental Exams		Anatomopathology
Patient	Sex	Age at Measles Exposure	Epidemiological Link	Measles Vaccine Doses and Age at Doses *	Age at Onset of SSPE Symptoms	SSPE Onset Symptoms	Clinical Stage **	Protein (mg/dL)	Cell mmc	Total IgG (mg/dL)	Link Index ^	Oligoclonal Bands ^^	EEG	MRI Brain: Areas of T2-Weighted Hyperintense Changes	Brain Biopsy
A	M	15 days	Mother	I (1 y); II (6 y)	3.9 y	Regressive behaviours, massive myoclonus, atonic seizures	2 A	29.3	0	19.9	10.78	Profile type 3 (more than 6 IgG-type bands in CSF)	Periodic bursts of high-voltage slow waves every 7 s	Frontal and parietal cortical and subcortical regions	Not performed
B	M	3 years	NA	I (1 y); II (7 y)	15 y	Focal and atonic seizures, massive myoclonus, cognitive impairment, ballistic movements	2 C (#)	29.3	3	4.66	1.32	Profile type 3 (more than 6 IgG-type bands in CSF)	Periodic bursts every 11 s with right-sided and temporo-parietal predominance	Centrum semiovale, internal capsule, corona radiata	Not performed
C	M	11 months	Mother	Not vaccinated	5.5 y	Cognitive impairment, focal epilepsy, acute deterioration with spastic tetraparesis, enteral feeding	3	29.4	2	21.5	7.08	Profile type 4	Periodic bursts with right-sided slow diffuse activity	Thalamic, mesencephalic, capsular, corpus callosum regions	Not performed
D	M	14 days	Mother	I (1.1 y)	3.9 y	Myoclonus, atonic seizures	2 B	23	1	12.9	2.25	Profile type 3 (at least 3 IgG-type bands in CSF)	Diffuse periodic bursts of high-voltage slow waves	Bilateral posterior parieto-occipital regions	Not performed

M = male; NA = information not available; EEG = electroencephalogram; MRI = magnetic resonance. * Doses are indicated in Roman numerals; age at dose of administration is expressed in years (y) in brackets. ** Clinical staging is defined according to Jabbour et al. [45]. ^ According to our laboratory, normal values for the Link Index are <0.7. ^^ Oligoclonal band patterns are defined as one of six classic types, according to Pinar et al. [65]. # In this case, the diagnosis was made 1 year after the onset of symptoms, at the age of 16, when the child was referred for the first time to our hospital.

**Table 5 viruses-14-00733-t005:** Laboratory findings.

	CSF Test at Diagnosis					Blood Test at Diagnosis			
Patient	IgG Anti-Measles AU/mL (ELISA)	IgM Anti-Measles	Total IgG mg/dL *	PRNT_80_	Measles Genome	IgG Anti-Measles AU/mL (ELISA)	IgM Anti Measles	PNRT_80_	Measles Genome
A	>300	No	19.9	320	Neg	>300	No	10,240	Neg
B	>300	No	6.6	320	Neg	>300	No	10,240	Neg
C	>300	No	21.5	160	Pos	>300	No	5120	Neg
D	>300	No	12.9	20	Neg	>300	No	80	Neg

* According to our laboratory, the normal values of total IgG in the cerebrospinal fluid (CSF) ranged from 0 to 4 mg/dL.

**Table 6 viruses-14-00733-t006:** Treatment details and follow-up in children with SSPE.

	Treatment			Follow-Up						
Patient	Duration of Intrathecal Interferon Therapy with 2b INTRON A	Co-Adjuvant Treatment	Other Treatments	6 Months		Last Follow-Up				
				Clinical Staging	EEG	MRI Brain	When	Re-Staging	Clinical	EEG
A	5-day escalation regimen *:one dose of 1,000,000 U/m^2^ weekly for 6 months; 1,000,000 U/m^2^ every 2 weeks for 6 months Total: 1 year	Oral inosiplex 100 mg/kg/day Total: 2.5 years	Antiepileptic drugs and motor rehabilitation	2B	Slow brain electrical activity, with a plurifocal peak and irregular short periods of voltage suppression	Progressive cortical atrophy, significant reduction in the corpus callosum	2.8 years later	2C from 2A; slow progression	Spastic tetraparesis, absent language, good understanding	Left-sided depressed activity without an ictal or periodic pattern
B	5-day escalation regimen *: one dose of 1,000,000 U/m^2^ weekly for 6 months; 250,000 U/day through a continuous intrathecal infusion pump for 2.5 years Total: 3 years	Oral inosiplex 100 mg/kg/day Total: 3.6 years	Antiepileptic drugs and motor rehabilitation	2C	Unorganised cerebral electrical activity and paroxysms of large slow waves at 2 c/s, with a periodic course and diffuse EEG expression	NA	4 years later	2C from 2C; stabilisation	Cognitive impairment, walking with support, ability to express primary needs	Poorly organised electrical activity without pathological potentials
C	5-day escalation regimen *: one dose of 1,000,000 U/m^2^ weekly for 12 months; 1,000,000 U/m^2^ monthly for 2 years Total: 3 years	Oral inosiplex 100 mg/kg/day Total: 3.2 years	Antiepileptic drugs and motor rehabilitation	3	Sequences of 2 c/s delta potentials with epileptiform elements in frontal–central regions bilaterally; no periodic pattern	Progressive atrophy, significant reduction in the corpus callosum, increased dilation of the ventricular system	7 years later	2D from 3; improvement	Spastic tetraparesis, improved environmental participation, oral feeding	Poorly organised electrical activity with ictal episodes associated with acoustic stimulation
D	5-day escalation regimen *: one dose of 1,000,000 U/m^2^ weekly for 3 months; after 3 months without therapy, the same therapy regimen repeated for another 6 months Total: 9 months	Oral ribavirin 20 mg/kg/day in two doses for 7 days; one dose of IgEV (1 g/kg) vitamin A 50.000 UI 2 times a day for 1 month	Antiepileptic drugs and motor rehabilitation	2B ** Progression	Slow cerebral activity with high-voltage theta waves (5 Hz) in centrum-temporo-parietal regions	Frontal right-sided cavitation area, resulting from the previous abscess; rapidly progressive global atrophy with extension of leukodystrophy; no recent tissue lesions	1 year later	2D from 2A; progression	Spastic tetraparesis, dystonia and tremors, enteral feeding	Poorly organised and asymmetrical electrical activity without an ictal pattern

* The 5-day escalation regimen included a daily dose of intrathecal 2b INTRON A, gradually escalated from 100,000 UI/m^2^ to 1,000,000 /m^2^. After 2 days’ rest, the dosage was 1,000,000 U/m^2^. Therapy was administered intrathecally via an Ommaya reservoir (or a similar device). ** The patient was referred to our hospital in December and administered therapy from December 2020 to February 2021. Intrathecal IFN was stopped precociously because of a cerebral abscess by methicillin-resistant *Staphylococcus aureus* undertreatment with antibiotics. Three months later, the same therapy regimen was re-administered because of a rapid neurological deterioration.

## Data Availability

The data presented in this study are available upon reasonable request from the corresponding author.

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
