# Peer review of "Subacute Sclerosing Panencephalitis in Children: The Archetype of Non-Vaccination"

_viruses, 2022, doi:10.3390/v14040733_

Round 1

Reviewer 1 Report

The manuscript provides a review of subacute sclerosing panencephalitis (SSPE) and describes the course of four cases. The report is interesting but could be improved.

  1. The writing needs substantial improvement.
  2. The authors need to review literature showing the SSPE has been associated with wild-type viruses.
  3. The results from the serologic testing are not convincing proof of intrathecal production of measles IgG. The authors should consider the approaches the following publications.
    1. https://pubmed.ncbi.nlm.nih.gov/1855284/
    2. https://www.sciencedirect.com/science/article/pii/S0165572807001269?via%3Dihub
    3. https://doi.org/10.1093/clinchem/37.7.1153
  4. Methods for serologic and RT-PCR testing should be included in the Methods and Materials section.
  5. The authors should specify the type of samples tested by RT-PCR in Table 4.
  6. No exposure information is provided for case B. This is the only case that was vaccinated before exposure, so the information regarding exposure is important.

Author Response

  • The writing needs substantial improvement. The work was revised by the journal's editing system (we attached the certificate). 
  • The authors need to review literature showing the SSPE has been associated with wild-type viruses. We discussed the role of wild type viruses and hypermutated variants in the paragraph “ Pathogenesis”
  • The results from the serologic testing are not convincing proof of intrathecal production of measles IgG. The authors should consider the approaches the following publications. In the discussion section we argued that detection of measles antibodies on CSF is required for diagnosis. We have also specified that the analysis methodology is decisive for discriminating between a pathogenetic production from an epiphenomenon.
  • Methods for serologic and RT-PCR testing should be included in the Methods and Materials section. The laboratory analysis methodology has been better specified in the methods section. We have reported the microbiological exams in a new table (table 5).
  • The authors should specify the type of samples tested by RT-PCR in Table 4. The sample was CSF.
  • No exposure information is provided for case B. This is the only case that was vaccinated before exposure, so the information regarding exposure is important. We have added details in the case description section (see table 5)

Reviewer 2 Report

Amodeo et al. report a series of 4 pediatric SSPE cases and a review of SSPE characteristics and case reports in the literature.

SSPE is a very serious condition and the number of cases has dramatically decreased in most regions, and therefore a case series could be of interest. However, this study has several limitations in terms of novelty and the low number of cases presented.

Most importantly, this manuscript is not written very well, and specifically:

1) The text needs a thorough English language review. Just in the first section of introduction: a) line 38 “is considered”; b) line 39 “COVID-19 pandemic” ; c) line 40 “Reports from England”; line 43 “…it can be predicted that there will be new outbreaks in the future …”; d) line 46 “…SSPE has the longest latency, from 1 month to 27 years,…”; e) line 50 “…has ever been proven” (and a reference is needed here); f) line 52 “…persistent virus replication”; g) line 53 “…which becomes clinical evident…”.

2) The text is not well organised and it needs some editing. For example: a) line 189, where is Table 4? Perhaps the authors mean Fig 4? b) line 189, I don’t think a reference to Table 3 is appropriate, as that table describes cases in the literature and not this case series; c) Line 290, there is no Table 5 in the manuscript. d) Table 3, the three Hosoya et al citations are not in the reference list; d) lines 128-131, the meaning of this paragraph is unclear and seems contradictory.

3) lines 165 and 220. What is the meaning of “natural” IFN-alpha?

4) The quality of the EEG figures is low and it is not possible to read the labels of the traces.

5) The Discussion repeats some of the information presented in the Introduction and in Results. I suggest to rewrite the Discussion so that it will just highlight the significance of this study compared to the previous literature.

Author Response

Most importantly, this manuscript is not written very well, and specifically:

  • The text needs a thorough English language review. Just in the first section of introduction: a) line 38 “is considered”; b) line 39 “COVID-19 pandemic” ; c) line 40 “Reports from England”; line 43 “…it can be predicted that there will be new outbreaks in the future …”; d) line 46 “…SSPE has the longest latency, from 1 month to 27 years,…”; e) line 50 “…has ever been proven” (and a reference is needed here); f) line 52 “…persistent virus replication”; g) line 53 “…which becomes clinical evident…”. the work was revised by the journal's editing system (we attached the certificate)..
  • The text is not well organised and it needs some editing. For example: a) line 189, where is Table 4? Perhaps the authors mean Fig 4? b) line 189, I don’t think a reference to Table 3 is appropriate, as that table describes cases in the literature and not this case series; c) Line 290, there is no Table 5 in the manuscript. d) Table 3, the three Hosoya et al citations are not in the reference list; d) lines 128-131, the meaning of this paragraph is unclear and seems contradictory. The tables have been reorganized, corrected and renumbered

3) lines 165 and 220. What is the meaning of “natural” IFN-alpha? we deleted the word “natural” as it was typed by mistake.

 4) The quality of the EEG figures is low and it is not possible to read the labels of the traces. we have improved the quality of the figures

5) The Discussion repeats some of the information presented in the Introduction and in Results. I suggest to rewrite the Discussion so that it will just highlight the significance of this study compared to the previous literature. we modified the discussion as suggested by the reviewer.

Reviewer 3 Report

The subject evoked by this article is of great importance in the context of the repeated resurgence of measles in all regions of the world as well as the interruption of a certain number of vaccination programs linked to the COVID crisis.

SSPE is a rare and very serious complication of measles that particularly affects children who have had a natural measles infection before the age of vaccination. The authors highlight this aspect very well throughout the article.

This pathology poses significant difficulties in care and treatment, but also in diagnosis.

Major points

  1. It would be important to summarize in the form of a table the data available at diagnosis for each of the 4 SSPE cases. L'évolution de chaque cas est présentée dans le tableau 4 mais les critères qui ont permis leur classification ainsi que le diagnostic de certitude sont essentiels and must be detailed.

  2. Laboratory diagnosis: which ELISA method was used to allow the detection or quantification of specific antibodies and in which type of sample. Was a homemade technique or a kit? What are the laboratory criteria for each case that led to the diagnosis.

    The laboratory criteria that allowed the diagnosis of each case should be mentioned in the table that will summarize the diagnosis

  3. What about the blood-brain barrier for each case at diagnosis. By what method its integrity was studied

  4. Further discussion of Case #3 is important. This child appears to have received a 1stdose of MMR vaccine before symptoms appear. The different hypotheses that would explain this situation must be developed (for example particular immune status, suspicion of a bad response after the first dose or measles during childhood not reported in the vaccination record, etc.)

  5. For patients B and C for whom the therapies have been stopped since 1 year and 4 years respectively, is there any information on the production of anti-MeV IgG long after the treatment is stopped?

  6. Were brain biopsies performed for some cases? If yes, was the detection by PCR for the measles virus carried out? Have any genetic mutations been identified?

Minor points :

line 35 : WHA is probably WHO

Line 335 : the word "in" is duplicated

Line 341 : the word « married » could be replaced by « associated »

Author Response

Major points

  • It would be important to summarize in the form of a table the data available at diagnosis for each of the 4 SSPE cases. L'évolution de chaque cas est présentée dans le tableau 4 mais les critères qui ont permis leur classification ainsi que le diagnostic de certitude sont essentiels and must be detailed.

 We reported these details in table 4 and 5.

  • Laboratory diagnosis: which ELISA method was used to allow the detection or quantification of specific antibodies and in which type of sample. Was a homemade technique or a kit? What are the laboratory criteria for each case that led to the diagnosis. We reported this data in a new table (table 5). The level of neutralizing antibody was in-vestigated by Plaque Reduction Neutralization Test (PRNT80)
  • The laboratory criteria that allowed the diagnosis of each case should be mentioned in the table that will summarize the diagnosis. These informations can be deduced from tables 4 and 5
  • What about the blood-brain barrier for each case at diagnosis. By what method its integrity was studied. The integrity of the blood brain barrier was studied through the determination of the oligoclonal bands on CSF and serum and with the link index (see table 4)
  • Further discussion of Case #3 is important. This child appears to have received a 1stdose of MMR vaccine before symptoms appear. The different hypotheses that would explain this situation must be developed (for example particular immune status, suspicion of a bad response after the first dose or measles during childhood not reported in the vaccination record, etc.) We specified in the case series paragraph that the patient did not have vaccination coverage (at the time he was exposed to the infection he had not completed the vaccination schedule).
  • For patients B and C for whom the therapies have been stopped since 1 year and 4 years respectively, is there any information on the production of anti-MeV IgG long after the treatment is stopped? Sorry, We do not have this data
  • Were brain biopsies performed for some cases? If yes, was the detection by PCR for the measles virus carried out? Have any genetic mutations been identified? brain biopsy was not required for diagnosis in any of our patients as they met the remaining major or minor criteria.

Minor points :

Ok line 35 : WHA is probably WHO

Ok Line 335 : the word "in" is duplicated

Ok  Line 341 : the word « married » could be replaced by « associated »

These typos have been corrected

Reviewer 4 Report

The paper presents the description of clinical features, diagnosis and treatment of four paediatric patients affected by Subacute sclerosing panencephalitis (SSPE), followed at the Bambino Gesù Childrens’ Hospital (Rome, Italy) in the last 10 years (2011–2021), who underwent a long-term combined intrathecal IFN-Α therapy. The Introduction presents an analysis of the pathogenesis, clinical features and treatments in patients with SSE.  

The paper’s quality can be increased by taking into account the following points:

  1. Line 189. Table 3 and Table 4. These tables are not included in the paper. The title of the tables are not indicated.
  2. Line 309. “A recent literature review…”. The reference for this review must be included in the paragraph.
  3. Line 319. “A systematic review…”. The reference for this review must be included in the paragraph.
  4. Lines 326-330. “In none of the SSPE series present in the literature we have data on possible post-exposure prophylaxis. In particular, we do not know whether previous SSPE cases have developed the disease despite post-exposure prophylaxis treatment. Data in this area would help to understand if this could be a valid strategy to reduce the risk of short- and long-term complications associated with measles.”

This paragraph must be reviewed because a study by Endo et al. assessing efficacy of postexposure prophylaxis against measles with immunoglobulin found a titer-dependent effect, with higher antimeasles titer providing the greatest protection. Children who did not develop disease received a mean dose of 10.9 IU/kg compared with 5.7 IU/kg for children in which postexposure prophylaxis with IGIM failed.

Endo A, Izumi H, Miyashita M, Taniguchi K, Okubo O, Harada K. Current efficacy of postexposure prophylaxis against measles with immunoglobulin. J Pediatr. 2001 Jun;138(6):926-8. doi: 10.1067/mpd.2001.113710. PMID: 11391343.

Author Response

The paper’s quality can be increased by taking into account the following points:

  1. Line 189. Table 3 and Table 4. These tables are not included in the paper. The title of the tables are not indicated. The tables have been reorganized, corrected and renumbered
  2. Line 309. “A recent literature review…”. We included the reference
  3. Line 319. “A systematic review…”. We included the reference
  4. Lines 326-330. “In none of the SSPE series present in the literature we have data on possible post-exposure prophylaxis. In particular, we do not know whether previous SSPE cases have developed the disease despite post-exposure prophylaxis treatment. Data in this area would help to understand if this could be a valid strategy to reduce the risk of short- and long-term complications associated with measles.” This paragraph must be reviewed because a study by Endo et al. assessing efficacy of postexposure prophylaxis against measles with immunoglobulin found a titer-dependent effect, with higher antimeasles titer providing the greatest protection. Children who did not develop disease received a mean dose of 10.9 IU/kg compared with 5.7 IU/kg for children in which postexposure prophylaxis with IGIM failed. We have added the results of the study by Endo et al.

Reviewer 5 Report

In this article, the authors reported four clinical cases of Subacute Sclerosing Panencephalitis (SSPE) in children. To date there is no standard protocol and doses recommended to treat these patients. According to the numerous studies they inoculated the patients with a combined intrathecal IFN-A therapy. The clinical manifestations of these four cases were followed up to 7 years. The incidence of SSPE increases a lot nowadays because of the lack of children vaccination and more case report articles describing long-term treatments are needed to help the community to decide on universal recommendations.

Major issues:

  1. Table 3 bis, Table 4 and Table 5 are missing in the manuscript I have received.
  2. The introduction should include more recent studies regarding MeV CNS infection as detailed below.
  3. The results section is not mentioned and is very short. The figures 1, 2 and 3 could also be part of the result section.

Minor issues and comments:

Introduction

  1. Line 35: specify that WHA is the world health assembly.
  2. Line 51: Authors should add one or two references that describe that SSPE incidence is higher with boys.
  3. Line 53: Authors claim that MeV CNS infection leads to a progressive destruction of the neurons. They should add some references. In addition, a lot of publications described the infection of oligodendrocytes and to a lower extent astrocytes and microglia. It can be worth to be mentioned.
  4. Line 54 (entire paragraph named Pathogenesis): I agree that in SSPE cases the M protein is often hypermutated. However, all recent publications focused on the mutations in the F and the H. The hyperfusogenic phenotype conferred by these mutations, especially in the F, has been extensively shown to be essential for viral dissemination in the CNS. A short paragraph should mention these mutations that can also explain the CNS invasion even in absence of the expression of MeV know receptors.
  5. Line 62: There are many other possibilities for the virus to disseminate within the brain, not only through synapses.
  6. Line 67-69: There is also an hypothesis suggesting that SSPE could affect people with a transient immunosuppression due to an exposure to another pathogen at the moment of MeV primary infection (PMID: 16260490).
  7. Lines 79 to 91: Neurokinin-1 is described as a receptor for the F protein and justifies even more that authors mentioned F and H mutations above in the introduction. Moreover, a lot of hypotheses have been made recently that can explain MeV dissemination withing the CNS. For example, cis-infection using CADM1 and CADM2 (PMID: 33910952 and 34788082) and nectin-elicited cytoplasm transfer (PMID : 31331966) but these hypothesis still have to be verified in a CNS context. More widely the ability of an hyperfusogenic F to promote fusion in absence of known receptor has been extensively shown to be determinant for MeV infection in the CNS (PMID : 25520515, 29298883, 30487282, 34061592).
  8. Figures 1 and 2: The text is really hard to read, even on the Tiff images, perhaps you can consider to rewrite the legend.
  9. Line 127 and all paragraph named treatment: I totally agree with the authors and the table helps.
  10. Line 145: This is a very interesting point that justifies why Authors did not use Ribavirin in this study. However, the Authors highlighted in table 3 and in “line 162” that Ribavirin is still a good option to be use in synergy with IFN-A too. The authors could insist in the discussion on why they did not try the Ribavirin on top of the treatment. That could help the readers to better understand the choice of the methodology.

Material and Methods

  1. Line 189: Table 3 and Table 4 are mentioned but I do not find them (it does not seem to be the same table 3 that already exists in the introduction). Indeed, it would help a lot to clarify the text.
  2. Line 188 and all paragraph named Case series: Regardless the lack of tables to show quickly the procedures, I found the methodology well explained, especially after considering the complexity of each case. Patient D is very different than patients A, B and C and authors could consider to treat this patient separately in the article.
  3. Vitamin A treatment is the only recommendation from the WHO to prevent further complications after MeV infection. Do patients A, C and D had a previous history of vitamin A treatment and do the authors have any comment about it?

Results

  1. I do not see any results section. It looks like it starts on line 233. The Figure 4 is clear. Perhaps the legend could include the colored box and the white box to facilitate the quick reading of the figure.
  2. The authors mentioned “years of therapy” and “year off-therapy”. This is absolutely correct but during the reading it was a little bit confusing.
  3. Do the authors have any data about the peripheral nervous system? Did they find viral antigens?

Discussion

  1. Line 288 to 290: Table 5 is missing. However, IFN-α treatments are known to be associated to several side effects. Do the Authors have an hypothesis that can explain why their patients did not develop interferonopathies for example? And how to deal with it afterward.
  2. Line 308: I totally agree, there is no specific recommendation from the WHO neither from any organism. Perhaps the authors can develop the fact that each patient have very different background and each case is unique. It seems that this is also the main reason why this is so difficult to treat SSPE patients. However, the long-term follow-up of the patients in this study is really useful.
  3. Line 336 to 346: This paragraph is interesting and emphasize the fact that the diagnosis is even more important for a disease like SSPE that needs to be treated as soon as possible. Do the authors have any suggestions to improve the early diagnosis? In the introduction the Authors mentioned from line 67 to line 78 the genetic predispositions that could increase the risk of contracting SSPE. An automatic screening of these genes for unvaccinated children that got infected with MeV before the age of 2 could be of interest.
  4. English: The language could be improved a little bit. In addition, there are some contractions in the text that sound familiar (line 43, 128) and some minor typo of remaining Italian words (SNC on line 152; Pazient in figure 4; PESS on line 335).

Author Response

Major issues:

1 Table 3 bis, Table 4 and Table 5 are missing in the manuscript I have received. The tables have been reorganized, corrected and renumbered

The introduction should include more recent studies regarding MeV CNS infection as detailed below. The introduction has been modified with more recent studies.

The results section is not mentioned and is very short. The result section has been reorganized. The figures 1, 2 and 3 could also be part of the result section. the results of the paper are summarized in tables 4 and 5.

Minor issues and comments:

Introduction

Line 35: specify that WHA is the world health assembly. it is a mistake and it is the WHO. We Changed it.

Line 51: Authors should add one or two references that describe that SSPE incidence is higher with boys. We added references.

Line 53: Authors claim that MeV CNS infection leads to a progressive destruction of the neurons. They should add some references. In addition, a lot of publications described the infection of oligodendrocytes and to a lower extent astrocytes and microglia. It can be worth to be mentioned.we have added the data and references suggested in the last paragraph of the introduction section. Further details have been addressed in the paragraph on pathogenesis.

Line 54 (entire paragraph named Pathogenesis): I agree that in SSPE cases the M protein is often hypermutated. However, all recent publications focused on the mutations in the F and the H. The hyperfusogenic phenotype conferred by these mutations, especially in the F, has been extensively shown to be essential for viral dissemination in the CNS. A short paragraph should mention these mutations that can also explain the CNS invasion even in absence of the expression of MeV know receptors.

We discussed the role of these mutations in the pathogenesis of the disease. In particular, we underlined the ability of  Mev to spread in the CNS despite the absence of receptors.

Line 62: There are many other possibilities for the virus to disseminate within the brain, not only through synapses. This topic has been discussed in the pathogenesis section.

Line 67-69: There is also an hypothesis suggesting that SSPE could affect people with a transient immunosuppression due to an exposure to another pathogen at the moment of MeV primary infection (PMID: 16260490). this suggestion has been included in the paragraph on pathogenesis.

Lines 79 to 91: Neurokinin-1 is described as a receptor for the F protein and justifies even more that authors mentioned F and H mutations above in the introduction. Moreover, a lot of hypotheses have been made recently that can explain MeV dissemination withing the CNS. For example, cis-infection using CADM1 and CADM2 (PMID: 33910952 and 34788082) and nectin-elicited cytoplasm transfer (PMID : 31331966) but these hypothesis still have to be verified in a CNS context. More widely the ability of an hyperfusogenic F to promote fusion in absence of known receptor has been extensively shown to be determinant for MeV infection in the CNS (PMID : 25520515, 29298883, 30487282, 34061592). In the pathogenesis section we have described the role of these proteins in the dissemination of the virus to brain tissue. In particular, we used the bibliographic suggestions indicated (thanks for the support).

Figures 1 and 2: The text is really hard to read, even on the Tiff images, perhaps you can consider to rewrite the legend. We tried to simplify the legends. Unfortunately there are quality criteria on the EEG data that we must respect, specifying the methods used.

Line 127 and all paragraph named treatment: I totally agree with the authors and the table helps. Thanks

Line 145: This is a very interesting point that justifies why Authors did not use Ribavirin in this study. However, the Authors highlighted in table 3 and in “line 162” that Ribavirin is still a good option to be use in synergy with IFN-A too. The authors could insist in the discussion on why they did not try the Ribavirin on top of the treatment. That could help the readers to better understand the choice of the methodology.

We have stressed in the discussion that although there are some data on efficacy (as reported in table 3)  in the use of intrathecal ribavirin, we have decided not to use this drug because the high therapeutic doses are very close to the toxic doses. Conversely, oral administration does not allow ribavirin to pass the blood brain barrier.

Material and Methods

Line 189: Table 3 and Table 4 are mentioned but I do not find them (it does not seem to be the same table 3 that already exists in the introduction). Indeed, it would help a lot to clarify the text. The tables have been reorganized, corrected and renumbered

Line 188 and all paragraph named Case series: Regardless the lack of tables to show quickly the procedures, I found the methodology well explained, especially after considering the complexity of each case. Patient D is very different than patients A, B and C and authors could consider to treat this patient separately in the article. The tables have been reorganized, corrected and renumbered. They help the reader to visualize the results.Patient D differs from the others for a complication of the intraventricular device while for the other aspects it can be superimposed. this is more evident in the tables (tables 4,5) summarizing the characteristics.

Vitamin A treatment is the only recommendation from the WHO to prevent further complications after MeV infection. Do patients A, C and D had a previous history of vitamin A treatment and do the authors have any comment about it? We added  our comments on the use and role of vitamin A in the SSPE.

Results

I do not see any results section. It looks like it starts on line 233. The Figure 4 is clear. Perhaps the legend could include the colored box and the white box to facilitate the quick reading of the figure. We have modified the legend of figure 4 and the result section.

The authors mentioned “years of therapy” and “year off-therapy”. This is absolutely correct but during the reading it was a little bit confusing. We have changed off therapy with years without therapy.

Do the authors have any data about the peripheral nervous system? Did they find viral antigens? Sorry, We have no data regarding this point.

Discussion

Line 288 to 290: Table 5 is missing. However, IFN-α treatments are known to be associated to several side effects. Do the Authors have an hypothesis that can explain why their patients did not develop interferonopathies for example? And how to deal with it afterward. Tables has been re-ordered. Our patients have expected side effects of interferon already described in the literature. In particular, fever, vomiting and lethargy. This aseptic is in the results section. What does the reviewer mean by interferonopathies? has some bibliographic references to suggest.

Line 308: I totally agree, there is no specific recommendation from the WHO neither from any organism. Perhaps the authors can develop the fact that each patient have very different background and each case is unique. It seems that this is also the main reason why this is so difficult to treat SSPE patients. However, the long-term follow-up of the patients in this study is really useful. Thanks we have underlined this comment in the text.

Line 336 to 346: This paragraph is interesting and emphasize the fact that the diagnosis is even more important for a disease like SSPE that needs to be treated as soon as possible. Do the authors have any suggestions to improve the early diagnosis? In the introduction the Authors mentioned from line 67 to line 78 the genetic predispositions that could increase the risk of contracting SSPE.  An automatic screening of these genes for unvaccinated children that got infected with MeV before the age of 2 could be of interest.

At the end of the discussion section we added that:

The identikit of the child with SSPE is that of a child who is always healthy, who has contracted or had exposure to a case of measles and who after a few years suddenly and progressively develops a neurological deterioration with atonic epileptic seizures. In these children, the EEG  with typical periodic complexes already adds a strong suspicion of SSPE. In these patients, performing lumbar puncture for the determination of Ig anti Mev in the CSF is mandatory. A possible target of future SPPE research would be to investigate whether children who contracted measles prior to vaccination could then be screened for risk of developing SSPE by studying susceptibility genes (MxA, interleukin-4 and interferon genes, Inf-α) we talked about above.

English: The language could be improved a little bit. In addition, there are some contractions in the text that sound familiar (line 43, 128) and some minor typo of remaining Italian words (SNC on line 152; Pazient in figure 4; PESS on line 335). We have revised the english by the editing system of the journal (we attached certificate)

Round 2

Reviewer 2 Report

Thanks to authors for revising extensively the manuscript and addressing satisfactorily all my previous comments.  I feel the new version is much improved in clarity and completeness. 

There are still a few minor typographical and language issues:

1) Line 42 – “…has further slowed vaccination…”

2) Line 260 – WHO identifies 23 (not 20) measles genotypes. The cited reference #16 is outdated and I suggest to substitute it for: “World Health Organization. 2003. Update of the nomenclature for describing the genetic characteristics of wild-type measles viruses: new genotypes and reference strains. Wkly. Epidemiol. Rec. 78:229-232”.

3) Line 997 – “Case description”

4) Line 1030 – “the viral genome was extracted”. The term “isolated” implies that the virus was cultured, which I assume is not the case here.  “analysed”: does this mean sequenced? Was the whole genome sequenced or only the N450 region? Please specify. Line 1057 – What does “superimposable” mean? Was the sequence identical, or similar, or belonging to the same genotype, named-strain, strain? Please specify.  What was the strain/genotype circulating in Italy? I think it was B3, but was there a specific named-strain? Please specify.

5) Line 1079 – What does “precocious dose of MMR” mean? Perhaps “early vaccination before 12 months of age” or “a dose of vaccine given prior to infection”? Please clarify

6) Line 1639 – “…other therapy schemes,”

7) Line 1172 – “…in the disease course…”

8) Line 1808 – “…a reduction in vaccination campaign compliance…”

9) Line 1882-1883 – “…in a clinically compatible context…”

10) Line 1814 – “…may be absent.”

Author Response

  1. The tapering and language errors have been corrected.
  2. The reference has been changed.
  3. The questions about analysis on viral genome have been clarified.
  4. Line 1030 regarding viral analysis. This poin has been moved and more clarified in the results section (red color)
  5. The statement on “precocious dose of MMR” has been modified.

Reviewer 3 Report

Version 2 of the article completes and significantly improves the different chapters.

The literature review is complete and the presentation of the cases followed has been improved.

The authors answered the questions

I recommend the publication of the article in the state

Author Response

Thanks

Reviewer 4 Report

The authors haver answered adeqautely to my review comments. The revision version of the paper is correct. 

Author Response

Thanks